# Assessment of preventive behavior and associated factors towards COVID-19 in Qellam Wallaga Zone, Oromia, Ethiopia: A community-based cross-sectional study

Birhanu Gutu[1][*], Genene Legese[1], Nigussie Fikadu[1][¤], Birhanu Kumela[1], Firafan Shuma[1], Wakgari Mosisa[1], Zelalem Regassa[2], Yoseph Shiferaw[3], Lata Tesfaye[4], Buli Yohannes[5], Kogila Palanimuthu[6], Zewudie Birhanu[7], Desalegn Shiferaw[1]

1 Department of Public Health, College of Medicine and Health Sciences, Dambi Dollo University, Dambi Dollo, Ethiopia, 2 Qellam Wallaga Zone Health Department, Oromia Regional Health Bureau, Dambi Dollo, Ethiopia, 3 Department of Chemistry, College of Natural and Computational Sciences, Dambi Dollo University, Dambi Dollo, Ethiopia, 4 Department of Physics, College of Natural and Computational Sciences, Dambi Dollo University, Dambi Dollo, Ethiopia, 5 Department of Biology, College of Natural and Computational Science, Dambi Dollo University, Dambi Dollo, Ethiopia, 6 Department of Nursing, College of Medicine and Health Sciences, Dambi Dollo University, Dambi Dollo, Ethiopia, 7 Deparment of Health, Behavior, and Society, Faculty of Public Health, Jimma University, Jimma, Ethiopia

☯ These authors contributed equally to this work.
¤ Current address: Department of Medicine, College of Health Sciences, Madda Walabu University, Bale Robe, Ethiopia
* birhanugutu@yahoo.com

**Data Availability Statement:** All relevant data are within the paper and its Supporting Information files.

## Abstract

### Background

The world is being challenged by the COVID-19 outbreak that resulted in a universal concern and economic hardship. It is a leading public health emergency across the globe in general and developing countries in particular. Strengthening good preventive behavior is the best way to tackle such pandemics.

### Objective

The study assessed preventive behavior and associated factors towards COVID-19 among residents of Qellam Wallaga Zone, Oromia Region, Ethiopia, 2020.

### Methods

A community-based cross-sectional study was conducted with a multistage sampling technique. Data were collected by interview and analyzed using SPSS version 23.0. Binomial logistic regression was used to test the association between the variables. An Adjusted Prevalence with 95% CI was used to express the associations and interpret the findings.

**Funding:** This study was funded by Dambi Dollo University. The university covered all costs related to the study design, data collection and analysis and preparation of the manuscript. BG received the fund and used for the intended purposes. BG, GL, NF, BK, FS, WM, YS, LT, BY, KP and DS: received salary from Dambi Dollo University. The fund has no specific grant number. The university website is: https://dadu.edu.et/.

**Competing interests:** The authors have declared that no competing interests exist.

**Abbreviations:** WHO, World Health Organization; COVID, CoronaVirus Disease.

## Results

Among 634 participants, 417(65.8%) were from urban residences, and 347 (54.7%) belongs to a female. Age ranges 18 years through 87 years. Only 68(10.7%) participants showed good preventive behavior for COVID-19. The majority of them (84.7%) perceived that the disease is very dangerous and 450(71.0%) of them believe that they are at high risk. More than 17% of the respondents have sufficient knowledge. Respondents with sufficient knowledge about COVID-19 were about 2 times more likely to exercise good preventive behavior compare to those with insufficient knowledge, [(APR: 2.1; 95% CI: [1.2, 3.9)]. The urban residents was 3.3 more than that of rural residents to practice good preventive behavior, [(APR: 3.3; 95% CI: [1.6, 6.4)]. Respondents who use social media as a source of information were more than 2 times more likely to have good preventive behavior compared to those who did not, [(APR: 2.3; 95% CI: [1.3, 3.4)].

## Conclusion

Adoptions of COVID-19 preventive behavior in the study population is very low. Due emphasis should be given to rural residents. Risk communication activities should be strengthened through effective community engagement to slow down and stop the transmssion of the disease in the community.

## 1. Introduction

Novel-coronavirus disease (COVID-19) is currently a public health emergency of international and local concerns [1]. The coronavirus belongs to a family of nidovirus that may cause various symptoms such as cough, fever, breathing difficulty, and lung infection. These viruses are common in animals worldwide, but very few had been known to affect humans [2].

Coronavirus disease was first found in Wuhan, China in late December 2019. The World Health Organization (WHO) used the term 2019 novel coronavirus to refer to a coronavirus [3]. The WHO announced that the official name of the 2019 novel coronavirus disease (COVID-19) to differentiate from other strains existing of the virus [4]. The current reference name for the virus is severe acute respiratory syndrome coronavirus 2 (SARS-CoV-2). It was reported that a cluster of patients with pneumonia of unknown cause was linked to a local Huanan South China Seafood Market in Wuhan, Hubei Province, China in December 2019 [5].

Since 31 December 2019, millions of people diagnosed with having the disease. Until August 09, 2020, more than 20 million confirmed cases, and more than 733,000 deaths were reported globally and the majority of the cases and deaths belong to the USA and Europe. The African continent is becoming the continent in which it is on spreading [6]. As of August 09, 2020 Ethiopia reported 22,253 confirmed cases and 390 confirmed deaths [7].

Personal protective practices such as rigorous hand hygiene, cough etiquette, and use of face masks, maintaining social distancing can contribute to reducing the spread of COVID-19. Ultra-careful hand-washing schemes, including washing of hands with soap and water for at least 40 seconds, or cleaning hands with alcohol-based solutions, recommended in all community settings in all transmission scenarios [8].

Evidence shows that effective preventive behavior depends on one's knowledge, attitude, and other socio-demographic characteristics. Most importantly arrays of floating rumors,

myths, and misperceptions regarding the spread, treatment, and nature of the virus can substantially affect people's knowledge, attitude, and adherence to standard COVID-19 protective measures. Thiswill in turn worsen the spread of the virus regardless of the beliefs [9, 10]. Thus, for effective risk communication programming, it is timely and imperative to assess people's level of understanding and adherence to standard precaution measures in Ethiopia and the study area. Therefore, this study aimed to assess the level of preventive behavior and associated factors among residents of Qellam Wallaga Zone of Oromia region, Ethiopia. The finding will help the local, regional and nationalplanners to devise effective strategies to prevent and control the spread of COVID-19 in the community.

## 2. Methods and materials

### 2.1. Study design and population

This study was conducted among residents of Qellam Wallaga Zone in the Oromia region. The capital of this Zone, Dambi Dollo, is located at about 637 km away from Addis Ababa, the capital city of Ethiopia. Qellam Wallaga Zone has a total population of about more than 1 million. Previously there was only one general hospital in this zone. Since 2019, three primary hospitals started to give service even if inadequate facility. The area is politically and physically disadvantaged where no research conducted and published till the current time.

### 2.2. Sample size determination and sampling procedures

**2.2.1. Sample size determination.** The sample size was calculated by using a single population proportion formula assuming that 50% preventive behavior for COVID-19 in the community; design effect of 1.5 and 10% non-response rate. This yield total sample size of 634 individuals.

**2.2.2. Sampling procedures.** The sampling method was multistage random sampling technique. In the first stage 3(25%), (Dambi Dollo, Sadi Chanka, and Dale Wabera), out of 12 districts were selected by simple random sampling method. Then, in each selected district, three Gandas (the smallest administrative unit in Oromia) were selected using a simple random sampling technique and within each selected Ganda 211.33 households were selected using a random sampling method. The total samples were allocated on an equal proportion to each selected Gandale. Households were the main sampling unit and eads of the households were primarily considered for interview from the selected households. For household heads that were not available, another person in the household who could properly respond to the questions were interviewed. The primary outcome of the study was preventive behavior towards COVID-19. The covariates of interest include socio-demographic characteristics, access to and source of information, knowledge, and perception towards this disease.

### 2.3. Variables and measurements

**2.3.1. Outcome variable.** *2.3.1.1. Preventive behavior for COVID-19.* Refers to the respondents' preventive activities that are recommended by WHO and MOH to reduce the spread of COVID-19 within the previous 2 days preceding the survey. It was measured by eight items with 5 point Likert scale response categories such as always, most of the time, sometimes, rarely, and never. The items include: wash hands regularly with soap and water; keep sanitizer based cleaner ready in case there is no water and soap for handwashing; don't touch mouth, nose, and eyes with the unclean hand; use a tissue or inner hand during coughing and sneezing; keep a distance of at least 2 meters from other people; cook meat and fish before eating; non-hand touch greeting, and don't travel crowded places. Finally, a composite variable termed comprehensive preventive behavior was produced by summing up the items as "good

preventive behavior" for those who practiced all the preventive measures most of the time and above and "poor preventive behavior" for the regression analysis and discussion.

### 2.3.2. Independent variables.

- **Socio-demographic factors:** These variables were used to assess the individual background information which includes sex, age, residence, marital status, religion, ethnicity, and educational status.

- **Information about COVID-19**: this includes access to and exposure to information, sources, and trust for the source of information about COVID-19.

- **Knowledge**: refers to the individual awareness and understanding about the disease. Four different dimensions were assessed including the general knowledge (three items), mode of transmissions (seven items), signs and symptoms (seven–items), and prevention methods of COVID-19 (seven-items), making a total of 24 items. Finally, researchers create a single dichotomous variable from the items depending on the score as insufficient knowledge for those who scored less than 80% and sufficient knowledge for those who scored 80% and above to use in the analysis.

- **Perception**: refers the respondents' insights about COVID-19 in general. This includes perception about the severity of the disease and perceived susceptibility for COVID-19.

## 2.4. Data collection instrument and technique

**2.4.1. Data collection tool.** The questionnaire was developed by the researchers as part of this study. The researchers reviewed relevant literature on COVID-19 including the Risk Communication & Community Engagement (RCCE) Action Plan Guidance for Covid-19 Preparedness & Response and the WHO advice on self-care recommendations as a base to develop the tool. Knowledge of the researchers about the local perceptions and practice were also helped to enrich the study tool. Then it was assessed by other public health experts to ensure the validity of its content. The questionnaire was translated into Afan Oromo and back-translated into English to ensure consistency of the translations. Exhaustive items were constructed to measure the intended variables. The tool has six parts. Part I deal with the socio-demographic characteristics of the study subjects and is assessed on eight items. Part II contains questions related to access to and source of information on COVID-19 which is measured by three multiple response questions. Part III relates to knowledge and contains six main questions some with multiple responses. Part IV comprehends 10 items relevant to perception about COVID-19. Part V measured the preventive behavior of the respondents with 8 items derived from recommendations by WHO and MOH on 5 Likert scale.

**2.4.2. Data collection technique.** The data were collected by face to face interview using structured questionnaire. Twelve (at least diploma holders) data collectors and three master holder supervisors were selected, trained, and participated in field works. Self-protective measures were taken during the interview and the fieldwork in general, by interviewers such as the use of face masks, maintaining physical distancing, and alcohol-based hand rub.

## 2.5. Data quality control

To assure the quality of data, the following measures were undertaken including pre-testing of the questionnaire, the final version of the questionnaire was translated into the respondents' language and intensive training was given to data collectors and supervisors and strict supervision was made.

## 2.6. Data analysis

The data were first checked manually for completeness and then coded; entered and cleaned using SPSS version 23. Descriptive analysis was used to describe the percentages and distributions of the respondents by socio-demographic and socio-economic characteristics. The main statistical method applied was a logistic regression with a binomial distribution and log link function. Variables which have a significant association with the dependent variable in the bivariate analyses at 0.25 were the primary target for multiple logistic regression model. Eligible variables were enter into the model to evaluate the methods that helps create a model that best fit the data and we found all the methods create similar quality model according the Hosmer and Limshow test of significance. Finally we use the stepwise forward (conditional) method to generate the model. Adjusted Prevalence ratio together with corresponding 95% confidence interval was used to interpret the findings "S1 Table".

## 2.7. Ethical clearance

The ethical clearance was conducted according to the higher education institutions' research ethics guideline. The ethical issues were checked and approved by Dambi Dollo University Institutional Ethical Review Committee. Then the official letter was received from the Dambi Dollo University Research and Technology Innovative directorate office. After having the official letter of the University, it was brought to the zonal health department, and selected district health office. A clear consent sheet was prepared and attached to the questionnaire for the data collectors to read for the participants just before the interview and written consent was obtained from individual participants. Confidentiality and privacy were maintained by excluding the name and ID of study participants from the questionnaire. Autonomy was maintained for both recruited & non recruited participants who are not willing to participate in the study was respected and they were not recruited in the study. All the enrolled participants were informed about the rights to withdraw from the study at any point in time. Justice was maintained by randomization to select the participants and veracity was maintained by truthfulness in each stage of the study.

## 3. Result

### 3.1. Socio-demographic characteristics of study participants

A total of 634 households from Sadi Chanka, Dale Wabara, and Dambi Dollo districts were included in the study with a response rate of 100%. The mean age of the respondents was 30.79 years with a standard deviation of 11.53 years. Three hundred and forty seven (54.7%) of respondents were females. Four hundred eighty nine (77.1%) have formal education. The majority, 604(95.3%) of the respondents were the Oromo ethnic group. Christians constitute more than seventy percent of the participants, 484(76.3%). Of the total respondents 485 (76.5%) of them are ever married; 242(38.2%) are farmers, and 417 (65.8%) reside in urban (Table 1).

**3.1.1. Access to information about COVID-19.** Most of the respondents 551(86.9%) already had information about how to protect themselves from the disease; 542(85.5%) knew the symptoms, and 516(81.4%) of them had information about methods of transmission (Table 2).

**3.1.2. Source of information about COVID-19.** Household respondents were asked about the source of information on the coronavirus with multiple response items; from the total 2113 positive responses, the majority of them 429 (20.3%) was radio followed by the TV which was 394(18.7%). The least response was of the traditional leaders and others 38(2%) (Table 3).

**Table 1. Socio-demographic characteristic of the study participants on assessment of preventive behavior and associated factors for covid-19 in Qellam Wallaga Zone, Oromia, Ethiopia, 2020.**

| Variables | Responses | Frequency (%) |
|---|---|---|
| Sex | Male | 287 (45.3) |
| | Female | 347 (54.7) |
| Age | < = 24 | 221 (34.9) |
| | 25–34 | 187 (29.5) |
| | 35–44 | 139 (21.9) |
| | 45+ | 87 (13.7) |
| Education | No formal education | 145 (22.9) |
| | Primary | 215 (33.9) |
| | Secondary | 189 (29.8) |
| | Tertiary | 85 (13.4) |
| Occupation | Farmer | 242 (38.2) |
| | Student | 128 (20.2) |
| | Merchant | 101 (15.9) |
| | Gov't employee | 86 (13.6) |
| | Daily labor | 77 (12.1) |
| Marital status | Unmarried | 149 (23.5) |
| | Ever married | 485 (76.5) |
| Ethnicity | Oromo | 604 (95.3) |
| | Amhara | 22 (3.5) |
| | Gurage | 7 (1.1) |
| | Other | 1 (0.2) |
| Religion | Orthodox | 70 (11.0) |
| | Protestant | 414 (65.3) |
| | Muslim | 147 (23.2) |
| | Other | 3 (0.5) |
| Residency | Urban | 417 (65.8) |
| | Rural | 217 (4.2) |

## 3.2. Knowledge of COVID-19

Four dimensions of knowledge about COVID-19 were assessed in this study; general, mode of transmission, signs and symptoms, and prevention methods. of the total study subjects 213 (33.6%), 87(13.7%), 33(5.2%) and 56(8.8%) of them gave correct response for the general, mode of transmission, signs and symptoms, and prevention methods dimensions respectively. Overall 130(20.5%) of the total respondents have sufficient knowledge while 504(79.5%) of them have relatively insufficient knowledge about COVID-19. Table 3 shows distribution of the knowledge about COVID-19 (Table 4).

**Table 2. Exposure to information on COVID-19 in Qellam Wallaga Zone, Oromia, Ethiopia, 2020.**

| What information do you ever heard about COVID-19? | Frequency (%) |
|---|---|
| How to protect onself | 551 (86.9) |
| Signs and Symptoms | 542 (85.5) |
| Transmission methods | 516 (81.4) |
| What to do if have symptoms | 276 (43.5) |
| Risks and complications | 193 (30.4) |
| Others* | 18 (2.8) |

**Table 3. Sources of information on COVID-19 in Qellam Wallaga Zone Oromia, Ethiopia, 2020.**

| Source of information: N = 634 | | N (%) |
|---|---|---|
| From radio | Yes | 429 (67.67%) |
| From TV | Yes | 394 (62.15%) |
| Health facility | Yes | 334 (52.68%) |
| From social media | Yes | 253(39.91%) |
| Religious leaders | Yes | 217(34.23%) |
| Any person from the community | Yes | 152 (23.97%) |
| Friends | Yes | 121 (19.09%) |
| Family member | Yes | 111 (17.51%) |
| Community leaders | Yes | 59 (9.31%) |
| Others | Yes | 22 (3.47%) |
| Traditional healers | Yes | 18 (2.84%) |

### 3.3. Perceptions about COVID-19

From the total 634 respondents, the majority of them (84.7%) perceived that the disease is very dangerous while about 4% of them thought the disease is not dangerous. Of total of 634 respondents, 450(71.0%) of them believe that they are at risk of acquiring the disease (COVID-19).

### 3.4. Preventive behaviors

Fig 1 indicates the distribution of preventive dimensions towards COVID-19. While the positive numbers on the right of the diagram indicate the frequency of respondents with preventive practice towards the disease, the negative sign before the numbers on the left side is to indicate the number of the respondents with behaviors that enhance the transmission of the disease. Among the top preventive behavior practiced always by the respondents was cooking meat before eating followed by wash hands regularly and non-hand touch greeting, 392(61.8), 327 (51.6%), and 322(50.8%) respectively (. Overall about 1 in ten respondents showed good preventive behavior while the remaining respondents have shown poor preventive behavior (.

### 3.5. Factors associated with COVID-19 preventive behavior

In this study three variables were found to independently associated with the respondents' preventive behavior for COVID-19. Respondents with sufficient knowledge were found to be more likely to adhere to good preventive behaviors compared with respondents who have insufficient knowledge, APR (95% C.I) 2.1 (1.2, 3.9). Being urban resident and using social media were also independently positively associated with good preventive behavior for COVID-19 (Table 5).

## 4. Discussion

The main aim of this study was to assess the preventive behavior of the respondents and associated factors. The level of preventive behavior on COVID-19 was low in our study. Of 634 respondents, only 68(10.7%) of them were practicing good preventive behavior. Evidence suggests that the spread of the disease has a direct relation relationship with the communities' adherence to the recommended practice. For example, the potential for pre-symptomatic transmission underpins the importance of adherence to recommended preventive behaviors [11]. However, the finding from our study indicates about more than 8 in 10 respondents have poor adherence to the recommended preventive behaviors. This figure is very alarming

Table 4. Knowledge about COVID-19 in Qellam Wallaga Zone, Oromia, Ethiopia, 2020.

| Dimensions | Items | Yes | No | I don't know |
|---|---|---|---|---|
| | | N(%) | N(%) | N(%) |
| General knowledge about COVID-19 | Does COVID-19 have a vaccine/treatment? | 29 (4.6) | 505 (79.7) | 100 (15.8) |
| | COVID-19 only affect elderly | 32 (5.0) | 552 (87.1) | 50 (7.9) |
| | One can completely cured from COVID-19 and be none-carrier | 269 (42.4) | 210 (33.1) | 155 (24.4) |
| Mode of transmission | Corona virus can be transmitted by blood transfusion | 100 (15.8) | 520 (82.0) | 14 (2.2) |
| | Coronavirus can be transmitted by droplet from an infected person | 416 (65.6) | 204 (32.2) | 14 (2.2) |
| | Coronavirus can be transmitted by direct contact with an infected person | 529 (83.4) | 91 (14.4) | 14 (2.2) |
| | Coronavirus can be transmitted by touching a contaminated object | 376 (59.3) | 244 (38.5) | 14 (2.2) |
| | Coronavirus can be transmitted by sexual intercourse | 171(27.0) | 449 (70.8) | 14 (2.2) |
| | Coronavirus can be transmitted by mosquito bites | 100 (15.8) | 520 (82.0) | 14 (2.2) |
| Signs and symptoms | Fever | 541 (85.3) | 63 (9.9) | 30 (4.7) |
| | Dry cough | 563 (88.8) | 41 (6.5) | 30 (4.7) |
| | Shortness of breath | 397 (62.6) | 207 (32.6) | 30 (4.7) |
| | Muscle pain | 228 (36.0) | 376 (59.3) | 30 (4.7) |
| | Headache | 409 (64.5) | 195 (30.8) | 30 (4.7) |
| | Diarrhea | 173 (27.3) | 431 (68.0) | 30 (4.7) |
| | Sore throat | 295 (46.5) | 309 (48.7) | 30 (4.7) |
| Prevention methods | Sleeping under mosquito net | 43 (6.8) | 585 (92.3) | 6 (0.9) |
| | Wash hands regularly with water and soap | 602 (95.0) | 26 (4.1) | 6 (0.9) |
| | Cover mouth and nose during coughing and sneezing | 392 (61.8) | 236 (37.2) | 6 (0.9) |
| | Avoid close contact with a person who has a cough and fever | 379 (59.8) | 249 (39.3) | 6 (0.9) |
| | Eliminate stagnant water | 130 (20.5) | 498 (78.5) | 6 (0.9) |
| | Cook meat and egg well before eat | 236 (37.2) | 392 (61.8) | 6 (0.9) |
| | Avoid contact with animals | 182 (28.7) | 446 (70.3) | 6 (0.9) |
| | Avoid hand contact/shaking | 540 (85.2) | 88 (13.9) | 6 (0.9) |
| Comprehensive Knowledge | Sufficient | 130(20.5) | | |
| | Insufficient | 504(79.5) | | |

towards the spread of the disease and necessitates further and rigorous enforcing and reinforcing efforts to strengthen the community's preventive behavior against the disease.

In this study urban residents were about three times more likely to have a good preventive behavior compared with rural residents, [(APR = 3.26; 95% CI: 1.65, 6.45)]. This difference may be due to the fact that urban residencies have accessibilities to different sources of information like TV and social media more frequently than rural residents. In addition to this, access to hygienic material and clean water may be a problem in rural parts in Ethiopia which can be another problem to adhere to the recommended behavior. According to a report from Central Statistics Agency on drinking water quality results from the 2016 Ethiopia Socioeconomic Survey, 74 percent of the population reported that it takes 30 minutes or less to collect drinking water [12]. Other study finding in Adama indicates only about 15% of beneficiaries could get 20 liters of water per day per capita in rural area [13]. This may indicates that accessing the rural area with necessary information and materials support is very important to control the spread of the disease.

The other variable having a significant association with the outcome variable is the respondent's level of knowledge about the COVID-19. Facts from different studies indicate knowledge is an important part in the formation of behavior [14]. For example, a lack of comprehensive knowledge has shown increased risks for another infectious disease like HIV

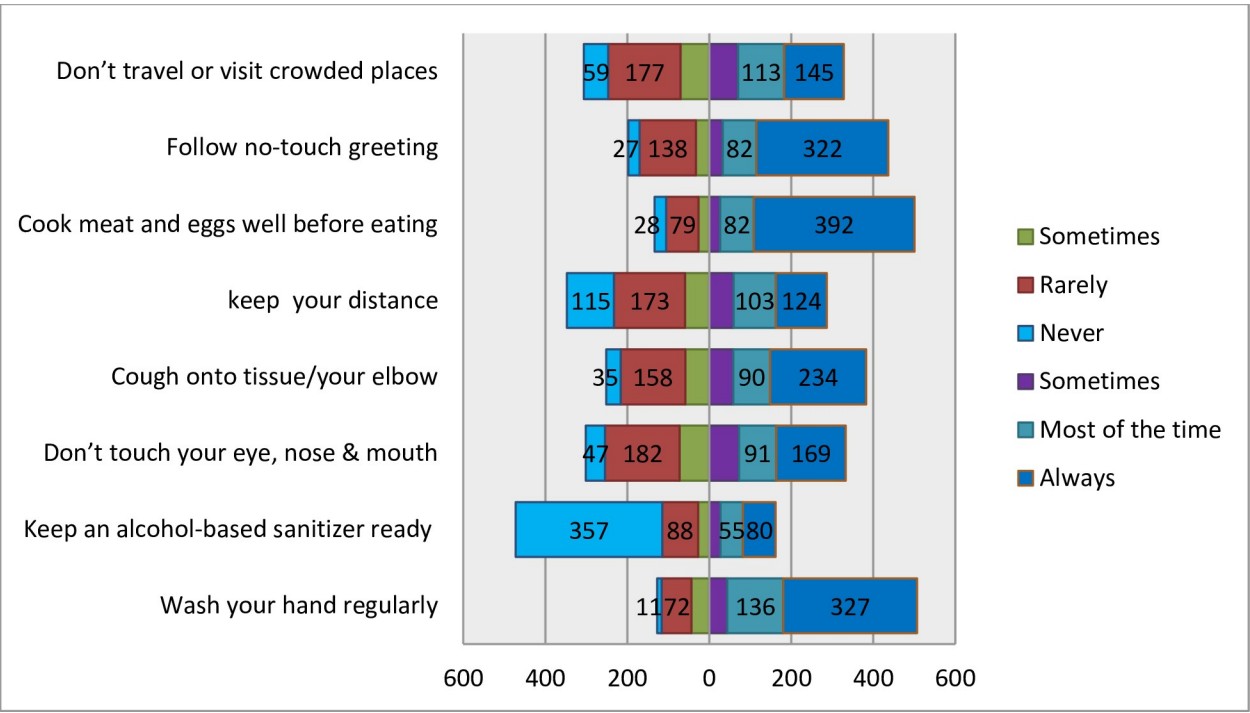

**Fig 1. Distribution of COVID-19 preventive behaviors, Qellam Wallaga Zone, Oromia, Ethiopia, 2020.**

while lower risk behavior is linked to better respondents' knowledge about HIV/AIDS [15–17]. A study was done in Ghana during the Ebola epidemic also shows that respondents of greater knowledge level were more likely engaged in preventive behavior [18]. According to the finding from our study, the respondents with sufficient knowledge were about two times more likely to practice the preventive behavior compared with those with insufficient knowledge, (APR = 2.140; 95% CI: 1.17, 3.93). This finding, in line with scientific reasons, clearly shows that as the community's level of knowledge increases their level of practicing the preventive behavior improves. This in turn may tells us that information dissemination and reinforcement is mandatory to improve the public knowledge about the disease towards protecting them.

**Table 5. Factors associated with COVID-19 preventive behavior in Qellam Wallaga Zone, Oromia, Ethiopia, 2020.**

| Variables | | Preventive behavior | | Unadjusted PR (95% CI) | APR (95% C.I) |
|---|---|---|---|---|---|
| | | Poor N (%) | Good N (%) | | |
| **Knowledge about COVID-19** | Sufficient | 109(17.2%) | 21(3.3%) | 1.9(1.1, 3.3) | 2.1 (1.2, 3.9) |
| | Insufficient | 457(72.1%) | 47(7.4%) | Ref | Ref |
| **Residence** | Urban | 360(56.8%) | 57(9.0%) | 2.9(1.5, 5.8) | 3.3 (1.6, 6.4) |
| | Rural | 206(32.5%) | 11(1.7%) | 1 | 1 |
| **Social media as trusted source of information** | Yes | 86(13.6%) | 20(3.2%) | 2.3(1.3, 4.1) | 2.3(1.3, 4.0) |
| | No | 480(75.7%) | 48(7.6%) | Ref | Ref |

Note: We executed binomial logistic regression by entering the entire variables of the study using similar entry method to evaluate in case there is/are confounders which affect the observed associations and the model gave the same result as of the previous binomial logistic regression indicating that the potential confounders are well controlled in our analysis.

In our study the use of social media as a source of information was also significantly associated with preventive behavior for COVID-19. Now days the world has become a global village where everyone can share information and learn about new things in the world being at his/her residence while social influences are primary factor in the adoption of health behaviors [19]. According to findings from a study conducted on effect of medias, media not only provides new information that persuades individuals to accept it but also can informs listeners about what others learn, thus facilitates preventive behavior [20]. In our study, those who use social media as their source of information on COVID-19 were twice more likely to practice COVID-19 preventive behavior compared with those who didn't, APR = 2.25; 95% CI: .258, 4.03). This may indicate that social media has significant power in the adoption of the recommended preventive behavior for COVID-19 and should be further strengthened.

## 5. Limitation of the study

Even if the study has its own strengths, it also has its own limitations. First, the preventive behavior of the respondents towards COVID-19 was classified in to good preventive behavior for individuals who scored 80% and above while poor preventive behavior for those respondents scored less than 80% in utilizing all forms of preventive activities on regular bases. This classification still doesn't assure that the respondents with good preventive behavior in this study will be at lesser risk as COVID-19 needs complete practice of the preventive actions on regular bases. Second, we use equal proportion of the sampling procedure and therefore, the absence of sampling weighting procedure in our analysis may introduced bias.

## 6. Conclusions

Collective preventive behavior was very low in this study. Of the total 634 respondents, only 68 (10.7%) of the them showed adherence to good preventive behavior for COVID-19. Urban residence, social media as source of trusted information, and sufficient knowledge about COVID-19 showed statisticallhy significant positive association with good preventive behavior. Due emphasis should be given to rural residents, improving social media coverage and utilization and means to improve knowledge about the disease to bring the intended level of preventive behavior in the community towards COVID-19.

## Supporting information

**S1 Table. Summary steps for the final model in determining predictors of preventive behavior towards COVID-19.**
(DOCX)

**S1 Data.**
(SAV)

**S1 Questionnaire. English version of the study questionnaire.**
(DOCX)

**S2 Questionnaire. Questionnaire Afan Oromo version.**
(DOCX)

## Acknowledgments

We would like to acknowledge Dambi Dollo University for giving us the opportunity and encouraging us to conduct this pandemic related investigation. Our sincere appreciation and

thanks go to all friends and colleagues who supported us in the process of conducting this research and writing the manuscript.

## Author Contributions

**Conceptualization:** Birhanu Gutu, Genene Legese, Nigussie Fikadu, Yoseph Shiferaw, Lata Tesfaye, Buli Yohannes, Kogila Palanimuthu, Zewudie Birhanu, Desalegn Shiferaw.

**Data curation:** Birhanu Gutu, Firafan Shuma, Wakgari Mosisa, Zelalem Regassa, Desalegn Shiferaw.

**Formal analysis:** Birhanu Gutu, Firafan Shuma, Zelalem Regassa, Desalegn Shiferaw.

**Funding acquisition:** Birhanu Gutu.

**Investigation:** Birhanu Gutu, Genene Legese, Firafan Shuma, Wakgari Mosisa, Zelalem Regassa, Desalegn Shiferaw.

**Methodology:** Birhanu Gutu, Zelalem Regassa, Lata Tesfaye, Buli Yohannes, Kogila Palanimuthu, Zewudie Birhanu, Desalegn Shiferaw.

**Project administration:** Birhanu Gutu, Desalegn Shiferaw.

**Supervision:** Birhanu Gutu.

**Validation:** Birhanu Gutu.

**Visualization:** Birhanu Gutu.

**Writing – original draft:** Birhanu Gutu, Zelalem Regassa, Desalegn Shiferaw.

**Writing – review & editing:** Birhanu Gutu, Genene Legese, Nigussie Fikadu, Birhanu Kumela, Firafan Shuma, Wakgari Mosisa, Zelalem Regassa, Yoseph Shiferaw, Lata Tesfaye, Buli Yohannes, Kogila Palanimuthu, Zewudie Birhanu, Desalegn Shiferaw.

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
