## [Decision Letter · Decision Letter 0]

11 Dec 2020

PONE-D-20-28112

ASSESSMENT OF PREVENTIVE BEHAVIOR AND ASSOCIATED FACTORS TOWARDS COVID-19 IN QELLEM WALLAGA ZONE OROMIA, ETHIOPIA: A COMMUNITY BASED CROSS-SECTIONAL STUDY

PLOS ONE

Dear Dr. Gutu,

Thank you for submitting your manuscript to PLOS ONE. After careful consideration, we feel that it has merit but does not fully meet PLOS ONE’s publication criteria as it currently stands. Therefore, we invite you to submit a revised version of the manuscript that addresses the points raised during the review process.

An expert in the field handled your manuscript, and we are thankful for their time and contributions. Although interest was found in your study, several major concerns and comments arose during review. Notably, there are questions about the experimental design, statistical analysis, and data presentation. Please address ALL of the reviewer's comments in your revised manuscript.

We look forward to receiving your revised manuscript.

Kind regards,

Frank T. Spradley

Academic Editor

PLOS ONE

"We would like to acknowledge Dambi Dollo University for financial aid."

"No, The funders had no role in study design, data collection and analysis, decision to publish, or preparation of the manuscript."

5. Please amend either the title on the online submission form (via Edit Submission) or the title in the manuscript so that they are identical.

Reviewers' comments:

Reviewer's Responses to Questions

**Comments to the Author**

1. Is the manuscript technically sound, and do the data support the conclusions?

Reviewer #1: Yes

2. Has the statistical analysis been performed appropriately and rigorously? 

Reviewer #1: No

3. Have the authors made all data underlying the findings in their manuscript fully available?

Reviewer #1: No

4. Is the manuscript presented in an intelligible fashion and written in standard English?

Reviewer #1: Yes

5. Review Comments to the Author

Reviewer #1: 50

change term "binary" logistic regression for "binomial" logistic regression

169

use logistic regression "with binomial distribution and log link"

50

change Odds Ratio for Prevalence Ratio. reason: It is a cross-sectional, not a case control study.

48

change multistage "systematic" sampling for multistage sampling

51

change test statistical significance for "to express the associations and"

116

remove word "systematic"

171

suggestion: detail the multiple stepwise process. specify if it was forward or backward. add a table to see the in which order the variables enter into the final model. review format in the output of summary(step.model$finalModel)in http://www.sthda.com/english/articles/37-model-selection-essentials-in-r/154-stepwise-regression-essentials-in-r/

165

observation: no details on how authors setup the survey sampling design prior to the logistic regression step. check: https://scialert.net/fulltext/?doi=ajms.2010.33.39

253

table 6 must specify the variables that were used to adjust the estimate. it also must show the unadjusted PR. put the outcome==1 in the right column (good prev behavior, in this case).

313

specify the reason of the restriction to make your data fully available

244

suggest to create figures to show the likert scale results as here https://www.r-graph-gallery.com/202-barplot-for-likert-type-items.html or here

https://cran.r-project.org/web/packages/sjPlot/vignettes/plot_likert_scales.html

6. PLOS authors have the option to publish the peer review history of their article (what does this mean?). If published, this will include your full peer review and any attached files.

Reviewer #1: No

---

## [Author Response · Author response to Decision Letter 0]

22 Jan 2021

Response to academic editor’s comments

Authors’ response: the manuscript revised according to PLOS ONE’s style requirements including the file naming

Authors’ response: we revised and put the detail of study tool and its preparation under the “Data collection tool” subtitle in the manuscript and included a copy of both the original language and English, as Supporting Information with the revised manuscript.

3. Financial disclosure statement 

Authors’ response: This study was funded by Dambi Dollo University. BG received the fund and used for the intended purposes. BG, GL, NF, BK, FS, WM, YS, LT, BY, KP and DS: received salary from Dambi Dollo University. The fund has no specific grant number. The university website is: http://www.dadu.edu.et.org. 

4. Data availability

Authors’ response: We uploaded the data file as Supporting Information with the revised manuscript

5. Please amend either the title on the online submission form (via Edit Submission) or the title in the manuscript so that they are identical.

Authors’ response: the title in the manuscript as well as on the online submission form are amended and identical

Response to reviewers' comments:

Comments to the Author

1. Is the manuscript technically sound, and do the data support the conclusions?

Reviewer #1: Yes 

Authors’ response: thank you

2. Has the statistical analysis been performed appropriately and rigorously? 

 Reviewer #1: No

Authors’ response: thank you for all the comments on statistical analysis, and data presentation. We amended all of them as suggested by the reviewer in our revised manuscript.

 3. Have the authors made all data underlying the findings in their manuscript fully available?

 Reviewer #1: No

Authors’ response: We uploaded our data as supporting information 

4. Is the manuscript presented in an intelligible fashion and written in standard English?

 Reviewer #1: Yes

Authors’ response: Thank you

5. Review Comments to the Author

 Reviewer #1: 

48 

Change multistage "systematic" sampling for multistage sampling

Authors’ response: accepted and change made

50 

Change term "binary" logistic regression for "binomial" logistic regression

Authors’ response: modified as per the comment

50 

Change Odds Ratio for Prevalence Ratio. Reason: It is a cross-sectional, not a case control study.

Authors’ response: Prevalence Ratio replaced the Odds Ratio throughout the entire manuscript

51 

Change test statistical significance for "to express the associations and"

Authors’ response: change made and "to express the associations and" is used instead of “test statistical significance for”

116 

Remove word "systematic" primarily 

Authors’ response: the word "systematic" is deleted 

169

 Use logistic regression "with binomial distribution and log link"

Authors’ response: change made as per the comment

165

 Observation: no details on how authors setup the survey sampling design prior to the logistic regression step. Check: https://scialert.net/fulltext/?doi=ajms.2010.33.39

Authors’ response: we accept the observation and thoroughly explained the sampling design in our manuscript as “sampling technique”; all the necessary steps we followed to reach the study unit (the households) and the respondents has discussed in the manuscript.

171

Suggestions: detail the multiple stepwise processes. Specify if it was forward or backward. add a table to see the in which order the variables enter into the final model. review format in the output of summary(step.model$finalModel)in http://www.sthda.com/english/articles/37-model-selection-essentials-in-r/154-stepwise-regression-essentials-in-r/

Authors’ response: Of course we planned to use multiple stepwise processes that would create the model that best fit our data based on the reality that we have large number of variables from our study. However, in our actual analysis all the methods produce similar model with similar Hosmer and Limshow’s model adequacy test result and as a reason we randomly used forward (conditional) method for variable entry in to the analysis. We also add a table titled “Variables in the Equation” from the output to see the in which order the variables enter into the final model as “supporting information” in our revised manuscript.

244 suggest to create figures to show the likert scale results as here https://www.r-graph-gallery.com/202-barplot-for-likert-type-items.html or here

https://cran.r-project.org/web/packages/sjPlot/vignettes/plot_likert_scales.html

Authors’ response: figures to show the likert scale results created as suggested

253 

table 6 must specify the variables that were used to adjust the estimate. it also must show the unadjusted PR. put the outcome==1 in the right column (good prev behavior, in this case).

Authors’ response: From the unadjusted analysis, we found only four variables which have a statistically significant association with the outcome variable namely residence, comprehensive knowledge, social media as trusted source of information, and traditional healers as trusted source of information. First, we used all these variables to determine the adjusted estimate. While the residence, knowledge and social media show statistically significant association in the model, traditional healers as trusted source has no statistically significant association with the outcome variable but still in the final model. Then we decided to use only the three variables to develop the model removing traditional healers as trusted source to compare the models’ adequacy in fitting the data. Excluding traditional healers as trusted source resulted in improved adequacy of the model from 0.126 to 0.282, according to Hosmer and Lemeshow Test of model adequacy. Therefore, because the variable (traditional healer as trusted source of information) has no statistically significant association adjusted to other variables in the model and removing the variable increases the model adequacy, we used only the three variables (residence, comprehensive knowledge, social media as trusted source) to adjust the estimate and generate the final model. And all the three variables remained in the model as you can see from table 6. We also revised table 6 to show the unadjusted PR, and put the outcome==1 in the right column (good prev behavior, in this case) as per the suggestions.

313 specify the reason of the restriction to make your data fully available

Authors’ response: we submitted the data file as supporting information with the revised manuscript

---

## [Decision Letter · Decision Letter 1]

23 Feb 2021

PONE-D-20-28112R1

Assessment of preventive behavior and associated factors towards COVID-19 in Qellam Wallaga Zone, Oromia, Ethiopia: A community-based cross-sectional study

PLOS ONE

Dear Dr. Gutu,

Thank you for submitting your manuscript to PLOS ONE. After careful consideration, we feel that it has merit but does not fully meet PLOS ONE’s publication criteria as it currently stands. Therefore, we invite you to submit a revised version of the manuscript that addresses the points raised during the review process.

We look forward to receiving your revised manuscript.

Kind regards,

Frank T. Spradley

Academic Editor

PLOS ONE

Reviewers' comments:

Reviewer's Responses to Questions

**Comments to the Author**

1. If the authors have adequately addressed your comments raised in a previous round of review and you feel that this manuscript is now acceptable for publication, you may indicate that here to bypass the “Comments to the Author” section, enter your conflict of interest statement in the “Confidential to Editor” section, and submit your "Accept" recommendation.

Reviewer #1: All comments have been addressed

2. Is the manuscript technically sound, and do the data support the conclusions?

Reviewer #1: Yes

3. Has the statistical analysis been performed appropriately and rigorously? 

Reviewer #1: I Don't Know

4. Have the authors made all data underlying the findings in their manuscript fully available?

Reviewer #1: Yes

5. Is the manuscript presented in an intelligible fashion and written in standard English?

Reviewer #1: Yes

6. Review Comments to the Author

Reviewer #1: Thank you for accepting the recommendations and congratulation for your efforts.

First, about the statistical analysis, I wonder if in your original procedures you applied a binomial logit regression (to estimate odds ratios) instead of a binomial log (to estimate prevalence ratios). My initial suggestion according to your study design was to update your analysis in case if it is required.

Also, I suggest to detail if you applied weights for survey analysis as explained here: https://dev.stats.idre.ucla.edu/stata/faq/can-i-use-working-weights-for-survey-analyses-in-spss/

Additionally, even though your statistical analysis followed a predictive procedure, have you evaluate how other covariates could affect the associations that you found? For example, the association between residence-behavior and knowledge-behavior could depend or be confounded by age, education or occupation. In case any of those apply as confounders, the reported estimates should be adjusted for this set of variables. This would not require to add more rows to the table 5, but specifing it as a caption or note at the end.

Related to this last point, I suggest to evaluate the consistency of the term "predictor" (as in subtitle 4.5) if you already applied "associated factor". Predictive modelling involve different objectives and procedures. Review Shmueli, Galit. "To explain or to predict?." Statistical science 25.3 (2010): 289-310. (https://doi.org/10.1214/10-STS330) and Chen, Lingxiao. "Overview of clinical prediction models." Annals of translational medicine 8.4 (2020). (https://doi.org/10.1214/10-STS330).

Furthermore, I recommend to add a limitation paragraph in the discussion section. There you could detail how your results may be affected by the validity or biases of the scale that you applied to define the outcome (preventive behavior), including references of previous experiences, if it apply. Also about the sampling procedure or the absence of sampling weighting procedure.

Lastly, I recommend to fix some writing typos like in lines 193, 194 or 223 from the "clean" copy of the manuscript. At line 188, it should end as "log link function". At table 4, you could add N(%) at the heading as in previous tables. At table 5, change the number "1" with "Ref." to detail the category of reference.

7. PLOS authors have the option to publish the peer review history of their article (what does this mean?). If published, this will include your full peer review and any attached files.

Reviewer #1: No

---

## [Author Response · Author response to Decision Letter 1]

10 Apr 2021

Comments accepted, amended and explained

---

## [Editor Report · Decision Letter 2]

20 Apr 2021

Assessment of preventive behavior and associated factors towards COVID-19 in Qellam Wallaga Zone, Oromia, Ethiopia: A community-based cross-sectional study

PONE-D-20-28112R2

Dear Dr. Gutu,

We’re pleased to inform you that your manuscript has been judged scientifically suitable for publication and will be formally accepted for publication once it meets all outstanding technical requirements.

Kind regards,

Frank T. Spradley

Academic Editor

PLOS ONE

---

## [Editor Report · Acceptance letter]

22 Apr 2021

PONE-D-20-28112R2 

Assessment of preventive behavior and associated factors towards COVID-19 in Qellam Wallaga Zone, Oromia, Ethiopia: A community-based cross-sectional study 

Dear Dr. Gutu:

I'm pleased to inform you that your manuscript has been deemed suitable for publication in PLOS ONE. Congratulations! Your manuscript is now with our production department. 

Kind regards, 

on behalf of

Dr. Frank T. Spradley 

Academic Editor

PLOS ONE